# Coronary CT Value in Quantitative Assessment of Intermediate Stenosis

**DOI:** 10.3390/medicina58070964

**Published:** 2022-07-20

**Authors:** Laura Zajančkauskienė, Laura Radionovaitė, Antanas Jankauskas, Audra Banišauskaitė, Gintarė Šakalytė

**Affiliations:** 1Department of Cardiology, Medical Academy, Lithuanian University of Health Sciences, LT-50161 Kaunas, Lithuania; laura.radionovaite@stud.lsmu.lt (L.R.); gintare.sakalyte@lsmuni.lt (G.Š.); 2Department of Cardiology, Kaunas Clinics, Hospital of Lithuanian University of Health Sciences, LT-50161 Kaunas, Lithuania; 3Department of Radiology, Kaunas Clinics, Hospital of Lithuanian University of Health Sciences, LT-50161 Kaunas, Lithuania; antanas.jankauskas@lsmuni.lt (A.J.); audra.banisauskaite@kaunoklinikos.lt (A.B.); 4Institute of Cardiology, Medical Academy, Lithuanian University of Health Sciences, LT-50166 Kaunas, Lithuania

**Keywords:** coronary CT angiography, cardiac CT, intermediate stenosis, quantitative assessment, coronary atherosclerotic plaque

## Abstract

*Background and Objectives*: Cardiac computed tomography angiography (CCTA) is an excellent non-invasive imaging tool to evaluate coronary arteries and exclude coronary artery disease (CAD). Managing intermediate coronary artery stenosis with negative or inconclusive functional tests is still a challenge. A regular stenosis evaluation together with high-risk plaque features, using semi-automated programs, are becoming promising tools. This case–control study was designed to evaluate the intermediate lesion features’ impact on CAD outcomes, using a semi-automated CCTA atherosclerotic plaque analysis program. *Materials and Methods*: We performed a single-center, prospective cohort study. A total of 133 patients with low to intermediate risk of CAD, older than 18 years with no previous history of CAD and good quality CCTA images were included in the study, and 194 intermediate stenosis (CAD-RADS 3) were analyzed. For more detailed morphological analysis, we used semi-automated CCTA-dedicated software. Enrolled patients were prospectively followed-up for 2 years. *Results*: Agatston score was significantly higher in the major adverse cardiovascular events (MACE) group (*p* = 0.025). Obstruction site analysis showed a significantly lower coronary artery remodeling index (RI) among patients with MACE (*p* = 0.037); nonetheless RI was negative in both groups. Plaque consistency analysis showed significantly bigger necrotic core area in the MACE group (*p* = 0.049). In addition, unadjusted multivariate analysis confirmed Agatston score and RI as significant MACE predictors. *Conclusions*: The Agatston score showes the total area of calcium deposits and higher values are linked to MACE. Higher plaque content of necrotic component is also associated with MACE. Additionally, negatively remodeled plaques are linked to MACE and could be a sign of advanced CAD. The Agatston score and RI are significant in risk stratification for the development of MACE.

## 1. Introduction

Cardiac computed tomography angiography (CCTA) is an excellent non-invasive imaging test to evaluate coronary arteries and exclude coronary artery disease (CAD) in patients with low to intermediate risk of CAD [1,2]. The main clinical advantage of CCTA is its negative predicting value [3]. According to the PROMISE trial, CCTA over weights stress testing alone in predicting adverse cardiac events among patients with stable chest pain [4]. However, the positive CCTA predicting value is lower, since interpretation is largely dependent on the reader’s clinical skills and experiences and can be overestimated [5].

With the assumption that most acute coronary syndromes occur in patients with non-obstructive CAD [2,4,5,6], novel and more accurate risk stratification models for non-obstructive CAD are necessary. Intermediate coronary artery stenosis is a topic of research in many of the latest studies as researchers have agreed that stenosis grade evaluation alone is insufficient. For example, one of the latest South Korean publications in this field has showed the importance of combined culprit lesions for future major adverse cardiovascular events (MACE) [7]. Invasive angiography additional techniques, such as fractional flow reserve, intravascular ultrasound (IVUS) or optical coherence tomography, are used to find the right decision with an intermediate lesion [8,9]. Even such a novel technique as deep learning has shown benefits in the evaluation of coronary lesions [10]. Nonetheless, the main factors for CAD management and prognosis continue to be atherosclerotic process extension and location as more data emerge supporting atherosclerotic plaque high-risk features [7,9,11]. Ordinary evaluations of the stenosis, together with high-risk plaque features for the most effective risk assessment using semi-automated programs, are becoming promising tools [12,13].

Managing intermediate coronary artery stenosis with negative or inconclusive functional tests is still a challenge. Intermediate coronary lesion is defined as 30% to 70% visual angiographic stenosis according to the literature [14]. Coronary Artery Disease Reporting and Data System (CAD-RADS) is a standardized scoring system providing simplified reports of CAD [15]. Current literature provides data verifying the reliability of CAD-RADS as different investigators have chosen the same scoring number [16]. With respect to clinical use and lower evaluation bias, we limited intermediate lesion definition to CAD-RADS 3 (50–69% stenosis). For more detailed morphological analysis looking for MACE predictors, we used semi-automated CCTA dedicated software (QAngio CT, Research Edition).

## 2. Materials and Methods

### 2.1. Study Population

A single-center, prospective cohort study was conducted at the Departments of Cardiology and Radiology, Lithuanian University of Health Sciences Kaunas Clinics, Kaunas, Lithuania. A total of 157 patients underwent clinically indicated CCTA and were diagnosed with one or more intermediate coronary artery (CA) stenosis between 1 January 2017 and 1 January 2020. Inclusion criteria were: older than 18 years, stable chest angina symptoms, low to intermediate ischemic heart disease probability, no previous history of CAD, normal systolic function by 2D echocardiography (>50%) and good quality of CCTA images. Only 133 of patients met all the criteria and were included in the study. Exclusion criteria were CAD diagnosis before enrollment visit (visually or clinically) and/or invasive treatment of CAD—stenting or bypass grafting. Four patients were lost during the follow-up (FU) period. A total number of 194 intermediate stenoses were analyzed. The intermediate lesion was defined as atherosclerotic plaque with 50–69% obstruction of the luminal area in a vessel with greater than a 1.5 mm diameter—CAD-RADS 3 in CCTA (patients with higher stenosis were excluded automatically). Enrolled patients were prospectively followed-up for 2 years (since the initial CCTA date), until 1 January 2022.

Phone interviews and medical records’ inspections were conducted to rule out MACE during the FU. Furthermore, patients were divided into two groups: MACE and absence of MACE group and later evaluated on occurrence of MACE during the 2-year period.

The study was approved by Kaunas Regional Biomedical Research Ethics Committee (project identification code BE-2-93, 14 October 2019) and performed in accordance with the criteria described in the declaration of Helsinki. Written informed consent was obtained from all subjects.

### 2.2. Image Acquisition

Baseline CCTA scans were performed with Toshiba 320-detector row CT scanner (Aquilion One; Toshiba Medical Systems, Nasu, Japan). Scan parameters were: detector collimation: 320 × 0.5 mm; tube current: 300–580 mA; tube voltage: 100–120 kV; gantry rotation time: 350 ms; and temporal resolution: 175 ms. Prospective electrocardiogram gating was used, covering 70–80% or the R-R interval. For images acquired at a heart rate (HR) of 65 beats per minute or slower, scanning was completed with a single R-R interval utilizing one segment. In patients with a HR greater than 65 beats per minute, data segments from two consecutive beats were used for multi-segment reconstruction with an improved temporal resolution of 87 milliseconds.

Premedication with beta-blockers was given if HR exceeded 60 beats per minute. For better image quality, sublingual nitroglycerine (0.4 mg per dose) was administered directly before CCTA. Each study participant underwent non-enhanced and contrast-enhanced CT. The non-enhanced scans were used for calcium calculation (Agatston score). CCTA was performed with 70–100 mL (vary according to patient weight) of iopromide containing contrast agent (Ultravist 370; Bayer Health Care; Leverkusen, Germany). Images were transferred and stored in the PACS system.

### 2.3. Imaging Analysis

Qualitative and quantitative plaque (limited to intermediate lesion) structure assessment was performed using a dedicated CCTA analysis program (QAngio CT, Research Edition, version 2.11.6.1, Medis Medical Imaging Systems, Leiden, The Netherlands) by two trained independent examiners as described earlier [9,12]. CCTA scans from the PACS system were transferred to the analysis program, which automatically located aorta and coronary arteries and provided 3D reconstruction images. After observing 3D reconstruction images, the affected coronary arteries or their branches were selected for further analysis. The program provided a longitudinal section of the selected artery that could be observed and assessed in various planes. The contours of the inner layer of the vessel—intima and outer layer—adventitia were automatically detected (when necessary, adjustments were made manually). Proximal and distal reference segments were also determined by the program automatically and corrected manually if necessary. After confirming defined vessel layers, the program automatically analyzed and measured lumen and vessel geometry and coronary plaque parameters. Furthermore, those parameters were evaluated at different levels: as whole lesion and the site of maximum obstruction.

### 2.4. Study End Points

The primary outcome was MACE. It was defined as all-cause death, revascularization of the intermediate target lesion, non-fatal myocardial infarction (MI) (as defined in ESC guidelines [17]) and cerebrovascular event (stroke, transient ischemic attack, intracranial hemorrhage). If there were two or more MACE at the same time or sequentially, it was counted as one event (the first one), and time-to-event duration was defined as duration from enrollment to the first event.

### 2.5. Statistical Analysis

Statistical analysis was performed using IBM SPSS statistics for Windows program (version 20.0. Armonk, NY, USA: IBM Corp). A *p*-value < 0.05 was statistically significant.

Descriptive and inferential statistical methods were used. Categorical variables were presented as frequencies and percentages, continuous variables—as mean (m) with (±) standard deviation (SD) or median with interquartile ranges (IQR). Patient demographics and plaque characteristics were compared between groups with and without MACE. For comparison of continuous variables, Wilcoxon–Mann–Whitney and Kruskal–Wallis tests were used. Categorical variables were compared using the Chi-square (χ_2_) test. After obtaining a statistically significant difference of the analyzed characteristic between the groups, its prognostic significance for the development of MACE was further calculated using multivariate Cox-Proportional hazard analysis.

## 3. Results

### 3.1. Baseline Characteristic

Mean age of the population was 65.3 ± 9.6; 59 (45.7%) patients were male. The most frequent localization of target lesion was left anterior descending artery (LAD) (59.9%). Demographic and clinical parameters compared between MACE and non-MACE groups are shown in Table 1.

The age and sex did not differ significantly between groups: in MACE and non-MACE groups mean age was almost identical at 65.3 ± 10.1 and 65.3 ± 9.5, respectively, (*p* = 0.909). There was no difference in lipid profile, major ischemic heart disease risk factors or creatinine level. Agatston score was significantly higher in the MACE group: 221.6 ± 204.1 compared to 149.1 ± 197.0 in the non-MACE group (*p* = 0.025).

The localization of intermediate CA stenosis also did not differ significantly between groups (*p* = 0.706). However, the most common localization was LAD in both groups (MACE—62.5% and non-MACE—59.8%).

The median FU duration was 25.3 (18.1–26.9) months with a median of 185.0 (73–573) days for time-to-event. Total MACE rate in the 2-year FU was 31.0%. The shortest time period to the event was 20 days, and the longest was 883 days. The most common MACE was revascularization (in all cases performing percutaneous coronary intervention) of the target stenosis (*n* = 33; 82.5%), followed by equally distributed nonfatal MI (*n* = 3; 7.5%) and stroke (*n* = 3; 7.5%) and one case of death (2.5%).

### 3.2. Quantitative CCTA Analysis

Semi-automated quantitative plaque analysis was performed on every intermediate lesion. Results were compared between MACE and MACE-free groups (Table 2).

On a lesion basis, no significant difference was detected between MACE and non-MACE groups.

Obstruction site analysis showed a significantly lower CA remodeling index (RI) among patients with MACE (*p* = 0.037); nonetheless RI remained negative (below 0.95) in both groups. The analysis of plaque consistency revealed that significantly bigger necrotic core area was related to MACE (*p* = 0.049) (Figure 1).

### 3.3. MACE Predictors

More than one-third of successfully followed patients developed MACE (31.0%) during the 2-year FU period. An unadjusted multivariate Cox-Proportional hazard analysis was performed with the previously mentioned predictors (CA remodeling index, relative and absolute necrotic core area). The significant MACE predictors were Agatston score and RI; thus a necrotic core area as an independent predictor did not show any prognostic value (Table 3).

## 4. Discussion

This case–control study was designed to evaluate intermediate lesion features (derived using a semi-automated CCTA atherosclerotic plaque analysis program) and their impact on CAD outcomes. Our results are quite heterogeneous but agree with recent scientific researches.

CA calcification is a well-established whole atherosclerotic process evaluation maker. Agatston score lets us evaluate the total area of calcium deposit. Several of the latest publications show a changing risk stratification position from a maximal CA obstruction point to a global atherosclerotic process evaluation [17,18,19]. Our study has proven that one of easiest way to evaluate calcium—Agatston score—is associated with future cardiovascular events: patients with a bigger score can experience MACE more frequently. Mortensen M.B. et al. demonstrated bigger contribution of a total plaque burden rather than stenosis grade alone for future cardiovascular event risk [18]. Deseive S. et al. have also reported higher total plaque volume as a risk factor for MACE in a 10-year period [20]. As our research was not designed to evaluate the whole atherosclerotic process impact on MACE occurrence, we can only make the assumption for future studies, explaining our research findings.

We have to point out that approximately 80% of all events in the MACE group was revascularization of the intermediate stenosis. This might have impact on the results as we compare MACE and non-MACE groups. According to the PROMISE trial, most MI and deaths occur in patients with non-obstructive CAD [4]. Consequently, our results raise questions about the rationality of invasive treatment and the bigger possibility of MI during FU if not for invasive treatment. Unfortunately, we did not assess functional test results that were performed before revascularization as the trial was not setup in this way.

The most unsuspected finding was from the maximal obstruction site RI analysis. Positive remodeling of atherosclerotic plaque is reported to associate with vulnerability [21]. In contrast, our study suggests that negatively remodeled plaques may be associated with increased risk of MACE. It can be explained by different plaque subtypes described by several former IVUS studies [22,23]. According to investigators, atherosclerotic plaque capability to remodel diminishes with time due to CAD progress. We did not differentiate premature CAD from advanced, but our population reflects primarily elderly patients (60 years and older). Following up on plaque pathogenesis [24] and premature CAD IVUS analysis made by Xie J. et al., the elderly are more likely to have diffused atherosclerotic process with negative remodeled and calcified CA [22]. In our study, most of the patients had negative remodeled coronary plaques, but MACE was more frequent in people with lower RI or, as we can say, with advanced CAD. In common with a higher Agatston score, lower RI has showed prognostic value for MACE accuracy. The Yu M. et al. research supports our findings as they have shown that lower RI was associated with higher CA stenosis [25]. This also corresponds to our previously discussed finding—higher Agatston score in MACE group.

Other significant findings rely on plaque composition. As previously established, low attenuation plaques (containing necrosis, fibrous, fibro–fatty elements) are linked to higher risk for developing acute coronary events [11,14,21,26,27]. Corresponding to the literature, our study has shown that a bigger relative part of the necrotic core area in atheroma (as a sign of an advanced phase of atherosclerotic process [22,28]) is related to increased frequency of MACE.

## 5. Conclusions

Semi-automated quantitative atherosclerotic plaque analysis is a useful tool in finding out vulnerable plaques that are detected with CCTA. The Agatston score reveals the total area of calcium deposits, and higher values are linked to more frequent cardiovascular events. The area of obstruction with a higher content of necrotic component is associated with more frequent cardiovascular events. Negatively remodeled plaques are linked to more frequent occurrence of MACE and could be a sign of advanced CAD. The Agatston score and RI are significant in risk stratification for the development of MACE.

## 6. Limitations

The study has some limitations. This was a retrospective study performed at a single center with a limited sample size. The latter one could have the biggest influence on results, in our opinion. Concerning the study population, it was limited by low to medium risk of CAD, which does not reflect all the spectrum of CAD risk stratification. Patients’ CAD risk factors and their control evaluation during the FU period, alongside medication adherence, could give additional information for the results. We also did not assess functional test results that could have validated the necessity of revascularization.

## Figures and Tables

**Figure 1 medicina-58-00964-f001:**
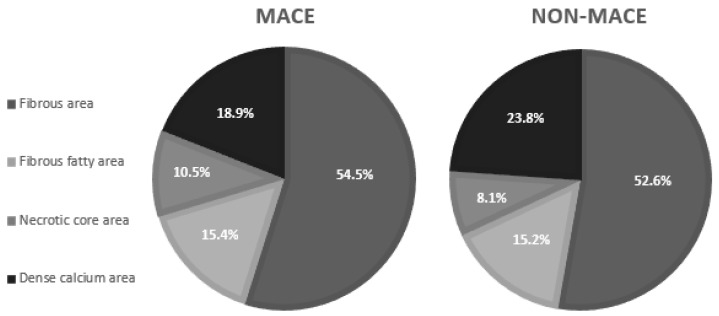
Maximal obstruction site cross-section plaque consistency analysis in MACE and non-MACE groups.

**Table 1 medicina-58-00964-t001:** Patients’ demographics.

Demographic Parameter	Total *n* = 129 (100%)	MACE	Non-MACE	*p* Value
		*n* = 40 (31.0%)	*n* = 89 (69.0%)	
Age, years	65.3 ± 9.6	65.3 ± 10.1	65.3 ± 9.5	0.909
Male (%)	59 (45.7)	20 (50.0)	39 (43.8)	0.515
Hypertension (%)	110 (85.3)	35 (87.5)	75 (84.3)	0.632
Diabetes mellitus (%)	21 (16.3)	8 (20.0)	13 (14.6)	0.443
Smoking (%)	25 (19.4)	10 (25.0)	15 (16.9)	0.279
Dyslipidemia (%)	95 (73.6)	32 (80.0)	63 (70.8)	0.272
Total cholesterol, mmol/L	6.1 ± 1.3	6.1 ± 1.5	6.1 ± 1.2	0.867
HDL, mmol/L	1.4 ± 0.3	1.4 ± 0.3	1.4 ± 0.3	0.659
LDL, mmol/L	3.9 ± 1.1	3.9 ± 1.2	3.9 ± 1.1	0.799
Triglyceride, mmol/L	1.7 ± 1.2	1.9 ± 1.6	1.6 ± 0.9	0.864
Creatinine, µmol/L	79.2 ± 16.3	81.8 ± 15.3	77.6 ± 16.8	0.109
Statin use (%)	93 (72.1)	31 (77.5)	62 (69.7)	0.521
Agatston score	171.6 ± 201.3	221.6 ± 204.1	149.1 ± 197.0	0.025

HDL—high density lipoprotein; LDL—low density lipoprotein.

**Table 2 medicina-58-00964-t002:** Quantitative CCTA analysis findings of target lesion.

	MACE *n* = 40 (31.0%)		Non-MACE *n* = 89 (69.0%)		*p* Value
	Mean ± SD	Median (IQR)	Mean ± SD	Median (IQR)	
Lesion area:					
Lesion length, mm	6.4 ± 4.3	5.1 (3.4–8.4)	6.3 ± 4.0	5.2 (3.6–8.0)	0.921
Vessel volume, mm^3^	75.9 ± 62.8	57.6 (35.5–94.8)	70.1 ± 45.6	60.2 (40.9–93.9)	0.943
Lumen volume, mm^3^	39.8 ± 38.9	31.0 (16.8–45.4)	36.3 ± 29.3	27.9 (17.8–43.7)	0.961
Plaque volume, mm^3^	37.3 ± 28.0	29.1 (17.3–55.8)	34.5 ± 23.3	29.6 (18.3–44.7)	0.815
Mean PB, %	44.6 ± 11.4	44.1 (37.0–52.3)	44.8 ± 12.9	43.3 (34.3–53.5)	0.867
Minimal plaque thickness, mm	0.03 ± 0.08	0.02 (0.00–0.03)	0.02 ± 0.04	0.01 (0.00–0.02)	0.122
Maximal plaque thickness, mm	1.6 ± 0.6	1.5 (1.2–2.2)	1.7 ± 0.6	1.7 (1.2–2.2)	0.539
Undefined plaque volume, mm3	0.18 ± 0.5	0.0 (0.0–0.5)	0.3 ± 1.4	0.0 (0.0–0.0)	0.885
TAG mean, HU/mm	−2.9 ± 18.9	−1.9 (−13.2–4.2)	−3.9 ± 14.9	−1.6 (−8.3–2.8)	0.848
TAG patch mean, HU/mm	−3.6 ± 16.5	−3.0 (−16.4–4.3)	−1.8 ± 17.3	−0.0 (−10.0–5.5)	0.450
Fibrous volume, mm^3^	18.1 ± 14.4	12.3 (8.6–28.6)	17.4 ± 12.8	14.4 (8.8–22.5)	0.982
Percent fibrous volume, %	54.1 ± 16.8	56.8 (43.6–67.3)	53.7 ± 20.2	53.8 (39.5–69.6)	0.855
Fibrous fatty volume, mm^3^	4.4 ± 5.5	2.5 (1.1–4.6)	3.9 ± 3.5	2.8 (1.5–4.9)	0.536
Percent fibrous fatty volume, %	15.3 ± 9.5	13.0 (6.8–23.0)	16.1 ± 12.1	14.0 (6.8–20.6)	0.887
Necrotic core volume, mm^3^	2.8 ± 4.3	1.1 (0.3–3.0)	2.8 ± 5.1	1.0 (0.4–2.7)	0.867
Percent necrotic core volume, %	11.3± 14.1	6.3 (1.3–17.7)	9.9 ± 13.1	0.4 (1.5–14.5)	0.587
Dense calcium volume, mm^3^	10.6 ± 13.0	4.6 (0.1–18.5)	9.5 ± 13.2	5.0 (0.1–15.0)	0.639
Percent dense calcium volume, %	18.5 ± 19.1	15.2 (0.3–30.4)	18.7 ± 18.8	15.2 (0.0–34.5)	0.955
Area of obstruction:					
Vessel wall area, mm^2^	10.1 ± 5.5	9.5 (6.2–13.6)	10.8 ± 4.7	10.1 (7.2–13.9)	0.347
Vessel wall diameter, mm	4.7 ± 8.9	3.5 (2.8–4.2)	3.5 ± 1.0	3.6 (3.0–4.2)	0.382
Eccentricity index	0.8 ± 0.2	0.9 (0.8–0.9)	0.8 ± 0.2	0.9 (0.8–1.0)	0.330
PB, %	58.0 ± 13.6	59.0 (47.8–66.0)	54.7 ± 16.4	53.8 (43.2–67.1)	0.226
Minimal plaque thickness, mm	0.2 ± 0.2	0.1 (0.1–0.3)	0.2 ± 0.2	0.1 (0.1–0.2)	0.491
Maximal plaque thickness, mm	1.3 ± 0.6	1.3 (0.9–1.8)	1.4 ± 0.6	1.3 (0.9–1.9)	0.671
Remodeling index	0.7 ± 0.2	0.7 (0.7–0.9)	0.8 ± 0.2	0.8 (0.7–1.0)	0.037
Lumen area stenosis, %	51.8 ± 10.9	51.8 (47.4–59.8)	50.3 ± 11.8	51.9 (42.5–57.3)	0.387
Lumen diameter stenosis, %	30.5 ± 9.9	30.8 (26.3–37.2)	29.4 ± 8.2	29.6 (24.8–34.1)	0.225
Undefined plaque area, mm^2^	0.06 ± 0.25	0.00 (0.00–0.00)	0.1 ± 0.3	0.0 (0.0–0.0)	0.459
Fibrous area, mm^2^	3.3 ± 1.4	3.0 (2.2–4.3)	3.3 ± 1.7	3.1 (2.2–4.3)	0.982
Percent fibrous area, %	54.5 ± 17.2	56.0 (43.1–66.0)	52.6 ± 21.0	52.0 (37.4–66.1)	0.518
Fibrous fatty area, mm^2^	0.8 ± 0.7	0.5 (0.4–1.0)	0.6 ± 0.4	0.6 (0.4–0.9)	0.747
Percent fibrous fatty area, %	15.4 ± 11.3	12.7 (5.7–22.9)	15.2 ± 13.7	11.2 (5.0–23.4)	0.504
Necrotic core area, mm^2^	0.5 ± 1.0	0.2 (0.1–0.4)	0.4 ± 0.6	0.1 (0.0–0.4)	0.047
Percent necrotic core area, %	10.5 ± 13.9	5.0 (1.0–15.2)	8.1 ± 13.6	2.3 (0.0–11.8)	0.038
Dense calcium area, mm^2^	2.0 ± 2.4	1.0 (0.0–3.7)	2.4 ± 2.7	1.4 (0.0–4.3)	0.623
Percent dense calcium area, %	18.9 ± 19.2	17.0 (0.0–34.2)	23.8 ± 22.6	20.9 (0.0–43.5)	0.441

TAG—transluminal volume gradient; HU—Hounsfield units, PB—plaque burden.

**Table 3 medicina-58-00964-t003:** Multivariate Cox-Proportional hazard analysis.

Variables	Hazard Ratio	95% CI	*p* Value
Agatston score	1.002	1.001–1.004	0.008
Remodeling index	0.171	0.030–0.979	0.047
Necrotic core area (mm^2^)	0.908	0.385–2.140	0.825
Percent necrotic core area (%)	0.996	0.951–1.104	0.878

CI–confidence interval.

## Data Availability

Not applicable.

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
