# Peer review of "Coronary CT Value in Quantitative Assessment of Intermediate Stenosis"

_medicina, 2022, doi:10.3390/medicina58070964_

Round 1

Reviewer 1 Report

Dear authors,

It has been a pleasure to go through this manuscript. Extremely well written, English used on an excellent level. However, I have the following comments in order to improve your work:

1.    The Introduction part could be more comprehensive. Recent works on diagnosis of intermediate stenosis of coronary arteries are closely related to the topic of the paper and should be reviewed.

2.    I would like you to add the inclusion criteria in paragraph 2.1. Study population.

  1. It would be benefic for this manuscript if you could add more information in the disscusion section. You should discuss your results with other comparable studies in this field.

4.    It is possible to add some CT images with the intermediate coronary artery stenosis? I'm sure it will make the manuscript even more compelling.

  1. Please add the abbreviations you have used in the manuscript.
  1.  

Author Response

1)            The Introduction part could be more comprehensive. Recent works on diagnosis of intermediate stenosis of coronary arteries are closely related to the topic of the paper and should be reviewed.

Authors’ response: We have revised The Introduction section and included additional information about the latest works in the field.

2)            I would like you to add the inclusion criteria in paragraph 2.1. Study population.

Authors’ response: We have added the inclusion criteria in paragraph 2.1.

3)            It would be benefic for this manuscript if you could add more information in the disscusion section. You should discuss your results with other comparable studies in this field.

Authors’ response: To our knowledge, it is the first time a research is performed with intermediate coronary artery stenosis CCTA with a 2-year follow-up period for major cardiovascular events. We revised literature data base in PubMed once again and added couple more comparable studies in Discussion part to make our manuscript topical.

4)            It is possible to add some CT images with the intermediate coronary artery stenosis? I'm sure it will make the manuscript even more compelling.

Authors’ response: With all regret we do not have the opportunity to retrieve CTA images from the analysis program (QAngio CT, Research Edition, version 2.11.6.1, Medis medical imaging systems, Leiden, the Netherlands) due to the end of the license.

5)            Please add the abbreviations you have used in the manuscript.

Authors’ response: We have added Abbreviations below Key words.

Reviewer 2 Report

The authors present the results of a single-center retrospective cohort case-control study that aimed to investigate the effect of intermediate stenoses on CAD outcomes. The article provides interesting information on this issue.

However, a critical limitation of the work in my opinion is the very small sample size (30 pts, 11 vs 19).

Using the standard features of the MedCalc software, I performed the following analysis. From Table 2 I selected the indicators for which there were statistically significant intergroup differences (p<0.05). For each indicator, the required sample size was calculated for Group 1 and Group 2 (taking into account the difference in mean values, SD in both groups, and the ratio of the number of patients in Group 1 to Group 2):

Lumen volume: Number of cases required in Group 1 – 32 (actual number 11), in Group 2 – 57 (actual number 19)

·      Remodeling index: Number of cases required in Group 1 – 8 (actual number 11), in Group 2 – 15 (actual number 19)

·       Lumen diameter stenosis: Number of cases required in Group 1 – 13 (actual number 11), in Group 2 – 23 (actual number 19)

·       Percent fibrous area: Number of cases required in Group 1 – 18 (actual number 11), in Group 2 – 32 (actual number 19)

Thus, in 3 cases out of 4, the statistical power of the study is insufficient to conclude about the differences that are actually present or absent. This makes the results extremely difficult for interpretation. The analysis should be performed on a sufficient number of patients.

Author Response

However, a critical limitation of the work in my opinion is the very small sample size (30 pts, 11 vs 19). Using the standard features of the MedCalc software, I performed the following analysis. From Table 2 I selected the indicators for which there were statistically significant intergroup differences (p<0.05). For each indicator, the required sample size was calculated for Group 1 and Group 2 (taking into account the difference in mean values, SD in both groups, and the ratio of the number of patients in Group 1 to Group 2):

Lumen volume: Number of cases required in Group 1 – 32 (actual number 11), in Group 2 – 57 (actual number 19);

Remodeling index: Number of cases required in Group 1 – 8 (actual number 11), in Group 2 – 15 (actual number 19);

Lumen diameter stenosis: Number of cases required in Group 1 – 13 (actual number 11), in Group 2 – 23 (actual number 19);

Percent fibrous area: Number of cases required in Group 1 – 18 (actual number 11), in Group 2 – 32 (actual number 19);

Thus, in 3 cases out of 4, the statistical power of the study is insufficient to conclude about the differences that are actually present or absent. This makes the results extremely difficult for interpretation. The analysis should be performed on a sufficient number of patients.

Authors’ response: During 2017/2018 research period in our medical center - Lithuanian University of Health Sciences Kaunas Clinics, only 450-500 cardiac CTA per year were performed. Intermediate CA stenosis (CAD-RADS 3) was an infrequent finding, only 6.7% of cardiac CTA ended with this diagnosis. We performed a relatively small sample size research with 2 year FU period for future studies. We agree on low statistical power but all tests where used that are suitable for small population and statistically correct.

Reviewer 3 Report

I would like to thank the authors on this very good research point.

I would like to add some comments/suggestions:

1) I think a major limitation of this study is the patients' control of risk factors and adherence to medical treatment. This could be a major confounder in the MACE results and is not incorporated in the study methods or mentioned in the limitation section.

2) The authors should give us some more data about what revascularization procedures were done for the patients. They said 16.7% in the MACE group had undergone revascularization, but did not mention how many in the non-MACE group. Also, in the discussion section, they said 2/3 of patients underwent revascularization, which is a completely different percent from that mentioned in the results section. Also, in the results section, they mention that the most common cause of MACE was revascularization. This is a very strong statement and needs more elaboration.

3) The authors mention it is a retrospective study. I think this study design is prospective.

4) Sample size: how did the authors calculate that sample size?

5) Page 6 line 191 and page 7 lines 203-207: the sentences are not completely understood (need restructuring to be more comprehended).

6) Some spelling mistakes e.g. compere (line 207) and differ (line 218).

Author Response

1)            I think a major limitation of this study is the patients' control of risk factors and adherence to medical treatment. This could be a major confounder in the MACE results and is not incorporated in the study methods or mentioned in the limitation section.

Authors’ response: Thank You for Your attentiveness. We discussed and added patients' control of risk factors and adherence to medical treatment in the Limitation section. We will take this into consideration for future studies.

2)            The authors should give us some more data about what revascularization procedures were done for the patients. They said 16.7% in the MACE group had undergone revascularization, but did not mention how many in the non-MACE group. Also, in the discussion section, they said 2/3 of patients underwent revascularization, which is a completely different percent from that mentioned in the results section. Also, in the results section, they mention that the most common cause of MACE was revascularization. This is a very strong statement and needs more elaboration.

Authors’ response: In section 3.1 Baseline characteristics we added additional information about revascularization as in all cases percutaneous coronary intervention was performed.

                16.7% revascularization in MACE group was derived percentage of all revascularization cases among all analyzed intermediate stenosis. We have discussed this number in researches group and agreed that it could be misleading, so we have changed MACE percentages in 3.1 Baseline characteristics showing what percentage every major cardiovascular event constitute in MASE group. Here are the numbers: revascularization (in all cases performing percutaneous coronary intervention) of the target stenosis (n=7; 63.6%), followed by equally distributed nonfatal MI (n=2; 18.2%) and stroke (n=2; 18.2%).

3)            The authors mention it is a retrospective study. I think this study design is prospective.

Authors’ response: We agree and have changed our study design to prospective.

4)            Sample size: how did the authors calculate that sample size?

Authors’ response: During 2017/2018 research period in our medical center - Lithuanian University of Health Sciences Kaunas Clinics, only 450-500 cardiac CTA per year were performed. Intermediate CA stenosis (CAD-RADS 3) was an infrequent finding, only 6.7% of cardiac CTA ended with this diagnosis. We performed a relatively small sample size research with 2 year FU period for future studies. We included as many patients as we could at that time.

5)            Page 6 line 191 and page 7 lines 203-207: the sentences are not completely understood (need restructuring to be more comprehended).

Authors’ response: Revised and restructured.

6)            Some spelling mistakes e.g. compere (line 207) and differ (line 218).

Authors’ response: Spelling revised, mistakes corrected.

Round 2

Reviewer 2 Report

I understand that enrollment of a sufficient number of patients meeting the study criteria is objectively difficult. It is true that the statistical methods used are applicable on small samples. However, each statistical procedure has its specific minimum required number of cases that make the results valid.

The statistical significance or insignificance of your differences cannot be considered convincing due to a critical underpower. In this regard, the findings are unlikely to be reproducible.

Author Response

 As our research was a continuous process, in the latest manuscript version we expanded our population and included new cases. This led to Results and Discussion graphs modification.

Round 3

Reviewer 2 Report

Dear authors, congratulations on a well-done and extensive work on the manuscript. I'm sure this work was necessary to improve the manuscript. 

The manuscript can be recommended for publication.